# PICH acts as a force-dependent nucleosome remodeler

Dian Spakman [1,4], Tinka V. M. Clement [1,4], Andreas S. Biebricher[1],
Graeme A. King[1,2], Manika I. Singh [3], Ian D. Hickson [3],
Erwin J. G. Peterman [1,5] ✉ & Gijs J. L. Wuite [1,5] ✉

In anaphase, any unresolved DNA entanglements between the segregating sister chromatids can give rise to chromatin bridges. To prevent genome instability, chromatin bridges must be resolved prior to cytokinesis. The SNF2 protein PICH has been proposed to play a direct role in this process through the remodeling of nucleosomes. However, direct evidence of nucleosome remodeling by PICH has remained elusive. Here, we present an in vitro single-molecule assay that mimics chromatin under tension, as is found in anaphase chromatin bridges. Applying a combination of dual-trap optical tweezers and fluorescence imaging of PICH and histones bound to a nucleosome-array construct, we show that PICH is a tension- and ATP-dependent nucleosome remodeler that facilitates nucleosome unwrapping and then subsequently slides remaining histones along the DNA. This work elucidates the role of PICH in chromatin-bridge dissolution, and might provide molecular insights into the mechanisms of related SNF2 proteins.

In eukaryotes, genomic processes such as DNA replication, transcription and repair are strongly influenced by chromatin architecture. The fundamental constituent of chromatin is the nucleosome, which comprises 147 base pairs of double stranded (ds)DNA wrapped ~1.7 times around a histone octamer, resulting in an inner and an outer turn of the DNA[1,2]. The canonical histone octamer consists of four homo-dimers of the histone proteins H2A, H2B, H3 and H4 and binds to the DNA via strong electrostatic interactions, occluding one complete face of the DNA helix from its environment[1]. Consequently, nucleosomes greatly affect how accessible the genome is for proteins to bind. It is therefore of great importance for genomic processes that the organization of nucleosomes is actively regulated. In vivo, this is enzymatically regulated by histone-modifying enzymes and ATP-dependent chromatin remodelers[3,4]. The latter can alter chromatin architecture through different mechanisms, including nucleosome sliding[5], nucleosome unwrapping[6], nucleosome eviction (i.e., removal of the histone octamer)[7] and exchange of histone dimers[8].

ATP-dependent chromatin remodelers belong to the SNF2 family of proteins and share a highly conserved ATPase domain[9], which facilitates binding to the minor groove of dsDNA[10]. Through a cycle of ATP binding and hydrolysis, the domain alternates between closed and open states, resulting in helical translocation of the protein along the DNA minor groove[5]. This translocation activity is thought to be the key mechanism of nucleosome remodeling. Although histone-DNA interactions could, in principle, be disrupted directly by collision between a moving translocase and a static nucleosome, in many cases translocation-induced remodeling is known to also involve the generation of torsional stress[10,11]. This usually occurs when the translocating remodeler encounters a 'roadblock' that hinders its helical translocation along the DNA. As a result, a DNA loop is extruded, which destabilizes nucleosomes indirectly by altering the DNA conformation. Roadblocks occur when the remodeler undergoes dimerization with another monomer[12], or in the form of other protein-protein or protein-nucleosome interactions[10]. To facilitate loop extrusion, many SNF2

[1]Department of Physics and Astronomy, and LaserLaB Amsterdam, Vrije Universiteit Amsterdam, De Boelelaan 1081, 1081 HV Amsterdam, The Netherlands. [2]Institute of Structural and Molecular Biology, University College London, Gower Street, London WC1E 6BT, UK. [3]Center for Chromosome Stability and Center for Healthy Aging, Department of Cellular and Molecular Medicine, University of Copenhagen, Blegdamsvej 3B, 2200 Copenhagen, Denmark. [4]These authors contributed equally: Dian Spakman, Tinka V. M. Clement. [5]These authors jointly supervised this work: Erwin J. G. Peterman, Gijs J. L. Wuite. ✉e-mail: e.j.g.peterman@vu.nl; g.j.l.wuite@vu.nl

proteins carry accessory domains that allow for protein-protein interactions and/or nucleosome binding[13]. In addition, these accessory domains are important for recruitment of other co-factors, such as actin-related proteins, specific histone variants or DNA-repair proteins[13].

While many SNF2 remodelers are primarily involved in the regulation of gene access and transcription[9], the role of the CSB/ERCC6 subfamily of remodelers in DNA-repair processes has recently received increased attention[14–25]. An intriguing example of such a protein is PICH, also known as ERCC6L, which plays a key role in the dissolution of ultrafine anaphase bridges (UFBs) during mitosis[16,17,25]. UFBs arise from sister chromatid entanglements and become apparent during anaphase when the mitotic spindle starts pulling the chromatids to opposite poles of the cell[25,26]. To ensure faithful sister chromatid segregation, it is essential that UFBs are resolved in a timely fashion[27,28]. PICH is thought to initiate the process of UFB dissolution both by stimulating topoisomerase TopoIIα[17,29] and by recruiting the BTRR dissolvasome[17,26,29–32]. Furthermore, bulk biochemistry and single-molecule studies have demonstrated that PICH displays many of the hallmark capabilities of chromatin remodelers, such as DNA translocation[16], DNA looping[33,34] and mono-nucleosome remodeling[30]. Although the DNA translocation and looping properties of PICH are similar to those of other SNF2 remodelers[16,33–36], its nucleosome remodeling activity has been reported to be orders of magnitude less efficient than that of other SNF2 remodelers[16].

Another difference from many other SNF2 proteins is that PICH mainly associates with the genome during mitosis, i.e., after breakdown of the nuclear envelope[25,29,37], when chromosomes are highly compacted. Curiously, while PICH exhibits a strong affinity for bare DNA[16], it lacks a nucleosome-binding domain[25], which suggests that it is unable to bind to mitotic chromosomes. It has therefore been hypothesized that the binding of PICH to chromatinized bridges (such as arising UFBs) only commences when chromatid entanglements result in a build-up of tension on the chromatin (via pulling forces of the mitotic spindle)[38–40]. This would result in local nucleosome unwrapping[16], exposing small regions of dsDNA to which PICH could bind, and thus act (indirectly) as a sensor for mitotic chromatin under tension. Subsequently, PICH could generate more bare dsDNA through tension-dependent nucleosome remodeling, which, in turn, would allow for a recruitment of repair complexes such as BTRR or TopoIIα[16]. This model would explain the weak chromatin-remodeling activity of PICH in ensemble biochemistry assays since the model implies nucleosome remodeling under tension, which cannot be emulated in ensemble chromatin-remodeling assays. Furthermore, the proposed mechanism of action of PICH would allow for specific remodeling of anaphase bridge-associated nucleosomes, while maintaining the integrity of non-entangled chromatin. However, to date, due to the limitations of ensemble chromatin remodeling assays, there is no direct experimental evidence to support this model.

Here, we employ a combination of dual-trap optical tweezers and confocal fluorescence microscopy to determine if PICH exhibits tension-dependent chromatin remodeling activity. To this end, we apply force to a dsDNA construct containing an array of nucleosomes to mimic chromatin under tension (such as is the case with anaphase bridges). In this way, we reveal that PICH can invade nucleosome arrays at tensions ≥3 pN, and can efficiently facilitate nucleosome unwrapping at tensions of ~5–10 pN in an ATP-dependent manner. Moreover, after nucleosome unwrapping has occurred, we observe that PICH can slide remaining histones along the DNA. We propose an updated model for the role of PICH in UFB dissolution, whereby PICH induces unwrapping of the inner turn of nucleosomes and subsequently slides histones to generate bare dsDNA. This allows additional PICH molecules to bind, providing an efficient substrate for TopoIIα and/or the BTRR dissolvasome to bind to, and thus promoting the timely dissolution of the UFB.

## Results

### PICH is able to invade nucleosome arrays at tensions ≥ 3 pN

We utilized combined dual-trap optical tweezers and confocal fluorescence microscopy to determine whether PICH exhibits force-dependent chromatin-remodeling activity. To this end, we reconstituted nucleosomes on a DNA construct containing an array of 12 repeats of the 601 nucleosome positioning sequence[41], each flanked by 25 base pairs of identical linker DNA[42]. The nucleosome array was located close to the center of a longer dsDNA molecule, such that the array was flanked by ~4 and ~4.5 kilo base pairs of dsDNA. These flanking DNA handles were biotinylated and thus allowed us to optically manipulate the construct using streptavidin-coated beads. To verify successful nucleosome positioning on our constructs, we first fluorescently labelled the nucleosome array with Anti-H3-Alexa647 antibodies. As shown in Fig. 1a, (i), this confirmed that reconstituted nucleosomes are located in the middle of the DNA construct. Next, to determine how PICH interacts with such a substrate, the nucleosome constructs were incubated (at 10 pN) in a microfluidic channel containing 25 nM PICH-eGFP and 2 mM ATP (Supplementary Fig. 1). As expected from the high affinity of PICH for dsDNA[16], PICH was found to readily bind to these constructs (Fig. 1a, (ii, iii)).

Nucleosomes positioned in an array are well-known to form higher-order chromatin structures, via nucleosome-nucleosome interactions between neighboring octamers, which can be disrupted by applying tensions of ≥3 pN[43,44]. When monitoring the locations of PICH bound to the nucleosome-array constructs at lower tensions (e.g., ~1 pN), PICH was found predominantly on the nucleosome-free DNA handles (Fig. 1b, (i)) and not the nucleosome array itself. This is consistent with PICH lacking a nucleosome-binding domain and thus a specific affinity for nucleosomes. In addition, we excluded the possibility that the relative absence of PICH in the nucleosome array at ~1 pN was merely due to the anti-H3-Alexa647 antibody blocking its access (Supplementary Fig. 2). Moreover, our findings suggest that, in the absence of tension, nucleosome-nucleosome interactions hinder the ability of PICH to access the regions of linker DNA (i.e., the DNA between neighboring nucleosomes). In contrast, when forces of ≥3 pN were applied to the construct, PICH was observed to also bind to the region of the construct containing the nucleosome array (Fig. 1a, b, (ii)). We attribute this observation to the release of nucleosome-nucleosome interactions[43,44], combined with a loss of nucleosome outer turns (of ~60 base pairs), which are well-known to start to unwrap at these forces (Fig. 1b, (ii), top)[38–40]. This results in increased accessibility of the linker DNA, allowing PICH to bind to these regions. The above observations support the previously proposed model in which PICH is only recruited to UFBs once the pulling forces from the mitotic spindle disrupt nucleosome-nucleosome interactions[16].

### PICH facilitates tension-induced nucleosome unwrapping using ATP hydrolysis

Given our observation that PICH can bind to linker DNA regions at forces ≥3 pN, we next examined whether PICH can alter tension-dependent unwrapping of nucleosomes. To this end, we monitored nucleosome unwrapping at different constant forces, using a force clamp ranging from 3 to 15 pN, in the absence and presence of PICH and ATP. At these elevated tensions, we assume that the outer turn is already unwrapped and that we mainly detect unwrapping of inner-turn nucleosomes[38–40]. We identified unwrapping events by monitoring step-wise increases in the end-to-end length of the nucleosome-array constructs (Fig. 2a, b). In the absence and presence of PICH and ATP, the average step size was found to be $26.4 \pm 0.2$ nm and $25.8 \pm 0.2$ nm respectively, which is consistent with the unwrapping of the remaining inner turn of ~80 base pairs (Supplementary Fig. 3)[2,42]. Notably, in the presence of PICH and ATP, transient and reversible shortening events of the DNA were also observed, for all forces tested (Fig. 2b). These events occurred throughout the constant-force

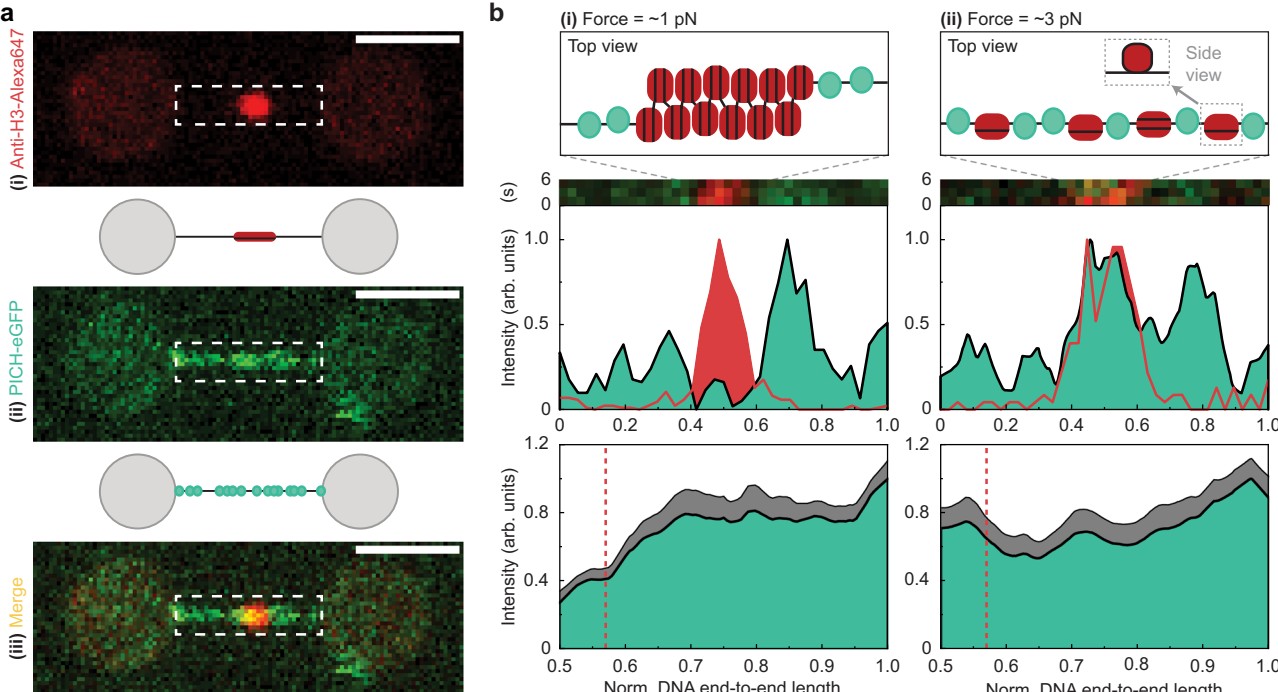

**Fig. 1 | PICH can bind to exposed dsDNA on nucleosome-array constructs under tension. a** Representative fluorescence images and schematic representations of a nucleosome-array construct with Anti-H3-Alexa647 antibody labelled histones, tethered between two optically-trapped beads, after 1-minute incubation with PICH-eGFP and ATP (at 10 pN). The Anti-H3-Alexa647 and PICH-eGFP signals and a composite of the two are displayed in (**i**), (**ii**) and (**iii**), respectively. The dashed white boxes highlight the region containing the nucleosome-array construct. Scale bars correspond to 2 μm. Similar PICH and histone distributions were observed on >10 independent nucleosome-array constructs. **b** *Top:* Schematic representations of PICH binding to a nucleosome array at -1 pN (**i**) and at -3 pN (**ii**). The DNA, histone octamers and PICH are depicted in black, red and green, respectively. *Middle:*

Representative kymographs (see Methods) and corresponding intensity profiles of PICH (green) and histones (red) after incubation of the construct with PICH-eGFP and ATP, at a tension of -1 pN (**i**) and -3 pN (**ii**). Note that in (**i**) and (**ii**) the histones were labelled with Anti-H3-Alexa647 and Atto-647N, respectively. The width of the kymographs corresponds to 2.9 μm. *Bottom:* Mean intensity profiles of PICH-eGFP visualized from the center of the DNA construct to each end at a tension of -1 pN (**i**) and -3 pN (**ii**). The dashed red lines indicate the area of nucleosome positioning sequences. $N = 28/14$ and $32/16$ (measurements/constructs) for -1 pN and -3 pN, respectively. Gray areas reflect +SEM. Source data are provided as a Source Data File.

measurements, independent of the presence of nucleosomes, and are indicative of the loop-extrusion activity of PICH that has previously been reported on bare DNA under tension by Bizard et al.[34]. This observation of loop extrusion, which is likely the result of oligomerization-induced roadblocks[12], is in agreement with the previously reported oligomerization of PICH[16] and the observation that the majority of PICH appeared to be present as oligomers during our constant-force measurements (Supplementary Fig. 4). Nevertheless, unwrapping events could still be determined, as the extruded loops persist only temporarily (Fig. 2b and Methods).

If PICH would assist nucleosome inner-turn unwrapping, we would expect that unwrapping occurs, on average, at lower forces in the presence of PICH and ATP than in their absence. To investigate this, we monitored the number of unwrapping events that occurred within 10 min under constant tension (Fig. 2a, b and Methods). The number of nucleosomes that had not yet unwrapped was then determined by stretching the construct and counting the number of force ruptures corresponding to the unwrapping of remaining nucleosomes (Fig. 2c)[2,42]. The unwrapped fraction was defined as the number of unwrapped nucleosomes (within the 10 min duration under constant tension, $N_{unwrapped}$) divided by the total amount of nucleosomes (unwrapped + remaining nucleosomes, $N_{total}$) (Eq. 1).

$$\alpha = \frac{N_{\text{unwrapped}}}{N_{\text{total}}} \tag{1}$$

As expected from previous studies[2,43,45,46], the fraction of unwrapped nucleosomes increased as a function of applied force.

More importantly, we observed larger unwrapped fractions in the presence of PICH and ATP than in its absence (Fig. 2d). To investigate whether the PICH-induced stimulation of inner-turn unwrapping of nucleosomes under tension is dependent on ATP hydrolysis, we repeated the above experiment at a force of 10 pN using AMP-PNP, a non-hydrolysable analog of ATP. Note that, in this case, the PICH concentration had to be increased by at least four-fold (100 nM instead of 25 nM) in order to achieve a similar coverage of PICH on the nucleosome-array construct as in the presence of ATP, indicating a stimulatory effect of ATP on PICH binding to DNA (Supplementary Fig. 5). Nevertheless, when using AMP-PNP instead of ATP, PICH did not increase the unwrapped fraction (+0.002, with a 95% confidence interval (CI) ranging from −0.16 to +0.17). In contrast, in the presence of ATP, a relative increase of +0.48 was observed (with 95% CI of +0.37 to +0.58) (Fig. 2d). Together, the above findings demonstrate that the PICH-induced stimulation of nucleosome inner-turn unwrapping is an active process and therefore rule out that the increase in the probability of tension-induced unwrapping is induced passively by binding, as has been recently demonstrated for HMGB proteins[46].

## ATP-dependent PICH activity reduces the lifetime of nucleosomes under tension

In order to assess the kinetics of nucleosome unwrapping under tension in the absence and presence of PICH and ATP, we determined the lifetime of inner-turn nucleosomes. We define this lifetime as the duration that an individual nucleosome on the array remains wrapped within a 10 min period at a constant force (Fig. 3a and Methods).

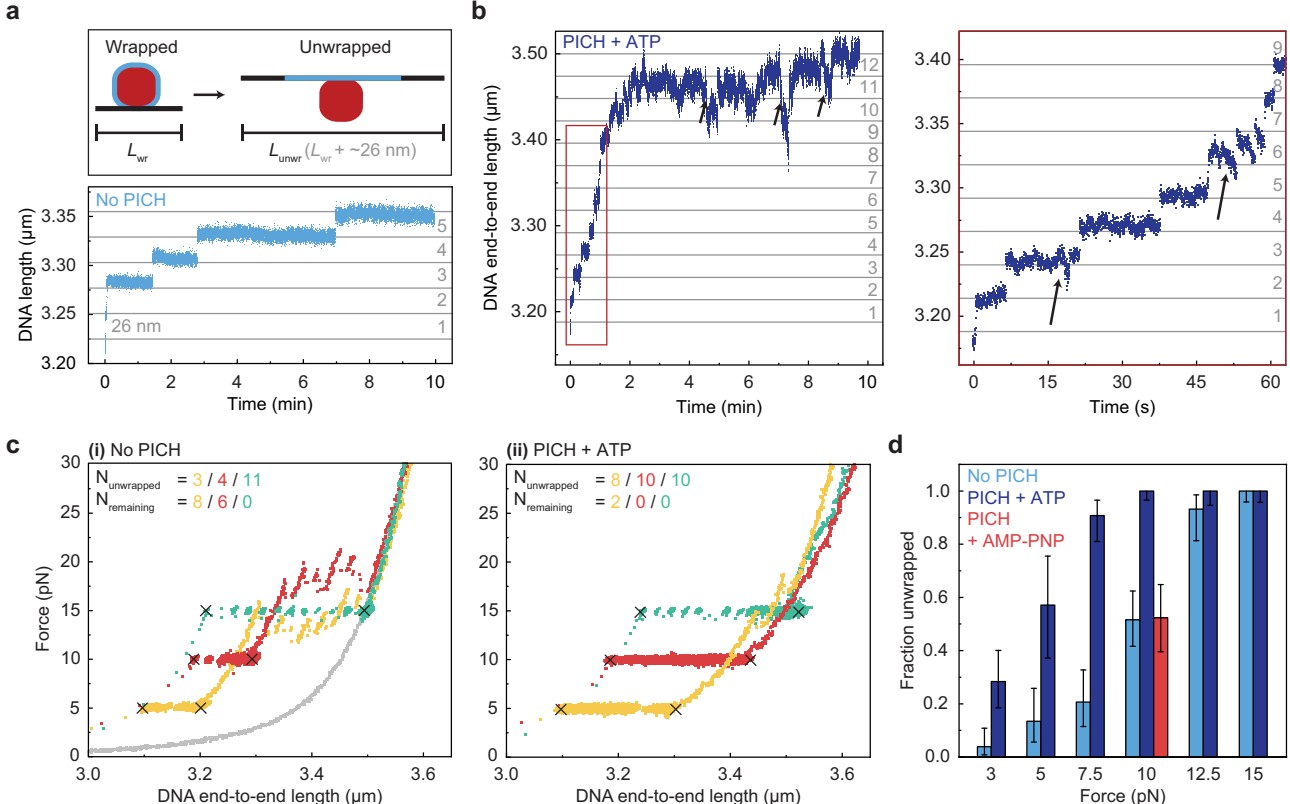

**Fig. 2 | PICH facilitates tension-induced nucleosome unwrapping using ATP hydrolysis. a** *Top:* Schematic representation of inner-turn nucleosome unwrapping, with DNA end-to-end length of the wrapped and unwrapped states as $L_{wr}$ and $L_{unwr}$, respectively. The histone octamer, inner-turn DNA, and outer-turn DNA are depicted in red, light blue and black, respectively. *Bottom:* Representative Distance-Time trace showing inner-turn nucleosome unwrapping events occurring over 10 min at 10 pN in the absence of PICH and ATP. Horizontal gray lines indicate 26 nm steps. The number of observed unwrapping events is indicated on the right. **b** *Left:* Representative Distance-Time trace of inner-turn nucleosome unwrapping occurring over 10 min at 10 pN in the presence of PICH and ATP. *Right:* A zoom-in of the constant-force measurement shown on the left. Horizontal gray lines indicate 26 nm steps. Black arrows indicate ATP-dependent loop-extrusion by PICH. The number of observed unwrapping events is indicated on the right side of each panel.

**c** Representative traces where a constant force of 5 pN (yellow), 10 pN (red), or 15 pN (green) was applied to nucleosome-array constructs for 10 min (indicated by black crosses), followed by stretching, to determine the number of nucleosomes that unwrapped and remained wrapped ($N_{unwrapped}$ and $N_{remaining}$, respectively) in the absence (**i**) and presence (**ii**) of PICH and ATP. The gray curve in (**i**) corresponds to the stretching trace of a construct when all nucleosomes had unwrapped. **d** Bar chart showing the measured fractions of all nucleosomes in a given condition that unwrapped during the 10 min constant-force measurements, in the absence of PICH (light blue, $N_{unwrapped} = 202$, $N_{total} = 421$) and the presence of PICH and ATP (purple, $N_{unwrapped} = 354$, $N_{total} = 425$). The measured unwrapped fraction at 10 pN in the presence of PICH and AMP-PNP is shown in red ($N_{unwrapped} = 34$, $N_{total} = 65$). Data are presented as unwrapped fractions ±95% CIs. Source data are provided as a Source Data File.

It should be noted that we were not able to determine reliable lifetimes at tensions below 10 pN due to the low fraction of unwrapped nucleosomes in the absence of PICH (Fig. 2d), and extensive loop-extrusion in the presence of PICH and ATP. We therefore only quantified and compared nucleosome lifetimes at 10, 12.5 and 15 pN (Fig. 3a). In total, 139 and 186 unwrapping events were collected in the absence and presence, respectively, of PICH and ATP, and these were used to calculate cumulative probability distributions (CPDs) (Fig. 3b, c). As mentioned above, in the absence of PICH, at 10 and 12.5 pN, not all nucleosomes unwrap within a 10 min period (Fig. 2d), resulting in remaining fractions of 0.48 and 0.07, respectively. We accounted for this by normalizing the distributions from the remaining fraction to 1.

As expected from the previously determined unwrapped fractions (Fig. 2d), this analysis confirmed that nucleosomes unwrap faster at increasing forces (Fig. 3b). Furthermore, at all tensions tested, the presence of PICH and ATP significantly reduced the nucleosome lifetimes (Fig. 3c). Remarkably, we found that the CPDs of the lifetimes do not follow a mono-exponential decay, but can best be described by biexponential decay functions, which we attribute to the presence of two distinct nucleosome populations; with 74 ± 1% belonging to a slower unwrapping population and the remainder to a faster population. This

behavior was independent of the presence of PICH and ATP (slower populations of 76 ± 1% in the absence and 72 ± 1% in the presence of PICH and ATP). The faster population unwraps more than two orders of magnitude quicker than the slower one at a force of 10 pN (Supplementary Fig. 6), indicating that it represents a subset of nucleosomes with a weaker histone-DNA interaction. We account for this observation by assuming the presence of tetrasomes in our assay. Tetrasomes comprise a nucleosome-intermediate, consisting of a H3/H4 tetramer (but no H2A/H2B dimers) and a single dsDNA wrap, and are known to be highly dynamic[47]. Furthermore, it is known that tetrasomes typically constitute ~20–40% of assembled histone complexes in reconstituted nucleosome arrays[48–50]. In line with these earlier observations, we speculate that our minor, faster population could reflect the unwrapping of tetrasomes.

Our aim was to investigate the interplay between PICH and canonical (i.e., properly assembled) nucleosomes. We therefore focused on the nucleosome lifetimes of the major and more stably bound population. The influence of PICH on these nucleosome lifetimes depended on the applied force and was most pronounced at 10 pN (the lowest force we were able to examine), with a 6.1-fold reduction of the mean lifetime (from 874 ± 51 to 143 ± 51 s). This decrease in lifetime was absent when AMP-PNP was used instead of

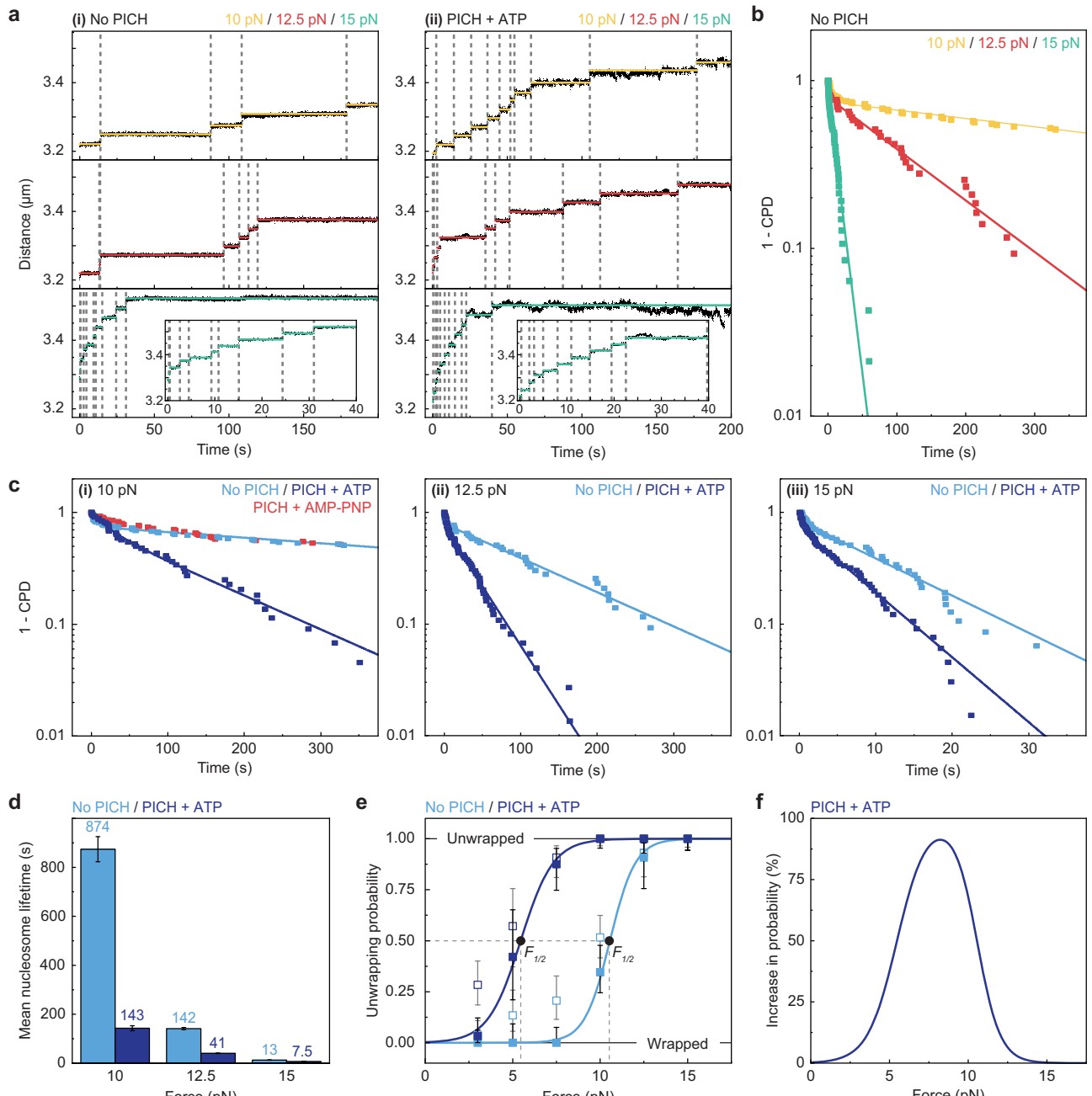

**Fig. 3 | PICH decreases nucleosome lifetimes and lowers the force required to achieve nucleosome unwrapping. a** Representative Distance-Time traces showing inner-turn-unwrapping events of nucleosomes at 10 pN, 12.5 pN and 15 pN, in the absence (**i**) and presence (**ii**) of PICH and ATP. Horizontal lines indicate segments detected by a step-fitting algorithm (see Methods). Unwrapping events are highlighted by the vertical dashed lines. **b** CPDs of nucleosome lifetimes in the absence of PICH at a constant force of 10, 12.5 or 15 pN, respectively. Solid lines display the corresponding bi-exponential decay fits. $N$ = 50, 41 and 48 nucleosome lifetimes for 10, 12.5 and 15 pN, respectively. **c** CPDs of the nucleosome lifetimes shown in (**b**) (light blue) and CPDs of nucleosome lifetimes in the presence of PICH and ATP (purple) at 10 pN (**i**); 12.5 pN (**ii**); and 15 pN (**iii**). Solid lines display the corresponding bi-exponential decay fits. $N$ = 45, 74 and 67 nucleosome lifetimes in the presence of PICH and ATP for 10, 12.5 and 15 pN, respectively. Red data points in (**i**) show the CPD of nucleosome lifetimes ($N$ = 34) at 10 pN in the presence of PICH and

AMP-PNP. **d** Bar chart showing the mean lifetimes of the major population of nucleosomes as determined by the fits shown in (**b** and **c**). Data are presented as mean lifetimes ± standard error of the fits. For $N$ of the total populations, see (**b** and **c**). **e** Fits of an Arrhenius-type equation (Eq. 2) to the tetrasome-corrected unwrapped fractions of nucleosomes in the absence (light blue) and presence (purple) of PICH and ATP (solid squares). The open squares reflect the measured unwrapped fractions, as shown in Fig. 2d, with $N_{total}$ = 421 and 425 nucleosomes in the absence and presence of PICH and ATP, respectively. Data are presented as unwrapped fractions ±95% CIs. The half forces ($F_{1/2}$) for inner-turn unwrapping derived from the fits are indicated. **f** A plot showing the percentage increase in the probability of inner-turn nucleosome unwrapping due to the presence of PICH and ATP. Data were obtained from a subtraction analysis of the sigmoidal curves displayed in (**e**). Source data are provided as a Source Data File.

ATP (Fig. 3c, (i)), consistent with our earlier finding that ATP hydrolysis is required to stimulate unwrapping. At higher tensions (12.5 and 15 pN), the effect of PICH on the mean nucleosome lifetime was less pronounced, with a 3.5- and 1.7-fold reduction, respectively (Fig. 3d).

The decreased effect of PICH on nucleosome lifetimes at higher tensions can be explained by the intrinsic effect force has on nucleosome lifetime: sufficiently high tensions (≥15 pN) result in fast nucleosome unwrapping, even in the absence of PICH (Fig. 3b, d)[2,45].

## PICH lowers the force required to achieve inner-turn unwrapping

The presence of a minor, faster population (assumed to be tetrasomes) has major implications for the previously determined fractions of unwrapped nucleosomes shown in Fig. 2d. As we are interested only in canonical nucleosomes, we corrected the unwrapped fractions such that they comprise only canonical nucleosomes (see Methods). This allowed us to extract additional quantitative information regarding inner-turn nucleosome unwrapping. Hence, we describe this process as a transition comprising two states (wrapped versus unwrapped). The force dependence of the unwrapping probability $P$ in the absence and presence of PICH and ATP can then be fitted with an Arrhenius-type equation (Eq. 2).

$$P(F) = \frac{1}{1 + e^{-\frac{\Delta x}{k_B T}(F - F_{1/2})}} \tag{2}$$

Here, $\Delta x$ reflects the effective distance to the transition state and $k_B T$ is the thermal energy, while $F$ and $F_{1/2}$ correspond to the applied and half force, respectively. The latter reflects the average force at which nucleosomes undergo inner-turn unwrapping.

When the above function was fitted to the corrected unwrapped fractions, it revealed that PICH lowers $F_{1/2}$ by about two-fold (from $10.6 \pm 1.7$ pN in the absence of PICH and ATP to $5.5 \pm 0.8$ pN in its presence, Fig. 3e). The value of $F_{1/2}$ in the absence of PICH is in good agreement with the ~8.9 pN reported previously[51]. Furthermore, the fit yields similar $\Delta x$ values of ~5 nm for the absence ($5.7 \pm 0.7$ nm) and presence ($4.4 \pm 0.5$ nm) of PICH and ATP. From these values the rate-limiting free energy barrier associated with inner-turn unwrapping, $\Delta G^{\ddagger}$, can be calculated[52]: $\Delta G^{\ddagger} = F_{1/2} \Delta x$, yielding ~$15 k_B T$ in the absence and ~$6 k_B T$ in the presence of PICH and ATP. Again, the $\Delta G^{\ddagger}$ in the absence of PICH is consistent with previously reported values of ~$15$–$16 \, k_B T$[51,53,54]. Our analysis thus reveals that, under our experimental conditions, the presence of PICH and ATP lowers the rate-limiting energy barrier associated with inner-turn unwrapping by ~9 $k_B T$; i.e., by more than 50%. The influence of PICH on the force-dependent probability of unwrapping can also be demonstrated graphically by subtracting the probabilities for the absence and presence of PICH (Fig. 3f). The force range within the full width at half maximum of this curve indicates that PICH facilitates unwrapping of nucleosomes most efficiently at forces between 5.3 and 10.7 pN. The peak of this curve additionally indicates that the effect of PICH on nucleosome unwrapping is strongest at forces of ~8.2 pN. At this tension, PICH increased the probability of unwrapping (within 10 min) by more than 90% (from ~4% to ~95% in the absence and presence of PICH and ATP, respectively). Taken together, these data suggest that, under our experimental conditions, PICH displays a maximum activity of force-induced nucleosome unwrapping in the force range of ~5–10 pN.

## PICH translocation can result in histone sliding along the DNA

We have shown previously that, in the absence of PICH, histones (at least histone H3) often remain bound to the DNA after force-induced nucleosome unwrapping has occurred[42]. In the present study, this binding stability of histones was confirmed by non-specifically labelling amine groups of the histone proteins bound to the nucleosome-array constructs with Atto-647N (see Methods). Although a decrease in the intensity of Atto-647N fluorescence was often observed during nucleosome unwrapping, substantial fluorescence signal persisted long after nucleosomes had unwrapped. To examine if PICH can affect the binding stability of histones, we next studied the interaction of PICH-eGFP with the Atto-647N-labelled nucleosome-array constructs. The correlation of fluorescence and nucleosome unwrapping data revealed that also in the presence of PICH and ATP, histones often remained bound to the DNA for at least several minutes after nucleosome unwrapping (Fig. 4a). This indicates that nucleosome

unwrapping does not result in complete histone eviction, even in the presence of PICH. Notably, these results suggest a difference in the underlying mechanisms of nucleosome remodeling between PICH and the structurally related SNF2 protein Rad54[9], for which direct evidence of histone eviction has been recently reported[55].

Despite the lack of complete histone eviction by PICH, we did observe an effect on the behavior of the remaining histones. Whereas in the absence of PICH we found that histones remained immobile after nucleosome unwrapping ($N = 21$ constructs, 10–12.5 pN), in the presence of PICH and ATP, displacements of histones were occasionally observed ($N = 3/18$). These histone displacements only occurred when there was a clear co-localization of histone and PICH fluorescence, and seemed to halt after collision with other bound PICH proteins (Fig. 4b). This suggests that PICH can slide histones along the DNA in an ATP-dependent manner and that histone displacements over larger distances might be possible with a less dense PICH coating (where the number of PICH proteins acting as road-blocks is minimized). We tested this idea by reducing the PICH coating ($7 \pm 1$ bound PICH proteins) and monitoring the interaction of PICH-eGFP with the Atto-647N-labelled histones at a constant tension of 12.5 pN. Under these low PICH-coverage conditions, continuous PICH-mediated histone sliding (typically over hundreds of nanometers) was observed in 65% of constructs ($N = 11/17$). In the majority of these cases, all nucleosomes had unwrapped before the onset of histone sliding ($N = 9/11$). In the other cases, histone sliding was observed when only a single inner-turn nucleosome remained wrapped. We also note that the observation of a histone sliding event does not necessarily mean that all histones present on the construct are being translocated by PICH. This is substantiated by the observation that often the Atto-647N signal was split: some of the labelled histones were displaced by PICH, while the rest remained stationary (Fig. 4c). Taken together, these results suggest that PICH only slides fully unwrapped histones.

Similar to the high-PICH coverage condition, histone sliding was only observed in cases where clear co-localization of the fluorescence signals of PICH and histones was observed ($N = 11$), which is consistent with the notion that SNF2 proteins typically slide nucleosomes through collision[5,7,55]. As determined from the fluorescence intensities, in the majority of cases histone sliding was driven by a single PICH protein that began translocating when bound between, or directly adjacent to, the labelled histone(s) (Fig. 4c). In other cases, the translocating PICH appeared as an oligomer, consisting of either two or three proteins, and sometimes, histone sliding resulted from PICH that was bound to the nucleosome-free DNA handle and translocated toward the histones (Fig. 4d). Dissociation of a PICH-histone complex from the DNA was observed only once in eleven trajectories during the tested timeframe. This is consistent with the strong affinity of PICH for DNA, as observed by Biebricher et al.[16]. Clear dissociation of PICH from a histone was not observed in any of the eleven trajectories. A more detailed analysis of the trajectories of histone sliding (Fig. 4e, (i) and Methods) revealed that most consisted of episodes of directed translocation (defined as ≥3 nm s$^{-1}$, $N = 28$ episodes) interspersed by pausing events (<3 nm s$^{-1}$, $N = 6$ episodes). In almost all cases, the histones and PICH co-translocated, with or without pauses, in a single direction until a roadblock (additional PICH or a bead) was reached (Fig. 4e (i)). The mean histone sliding velocity was found to be $10.1 \pm 0.8$ nm s$^{-1}$. This velocity is similar to that reported for PICH alone ($9.8 \pm 0.2$ nm s$^{-1}$)[16], suggesting that the translocation rate of PICH is not reduced by sliding histone(s). Moreover, we detected no significant difference between the velocities of PICH-histone trajectories involving PICH monomers and oligomers, respectively ($9.2 \pm 0.7$ versus $11.9 \pm 2.0$ nm s$^{-1}$, Fig. 4e, (ii)). Together, these results indicate that after nucleosome unwrapping, ATP-dependent translocation of PICH monomers and oligomers can result in sliding of histones along the DNA, without affecting the translocation rate of PICH.

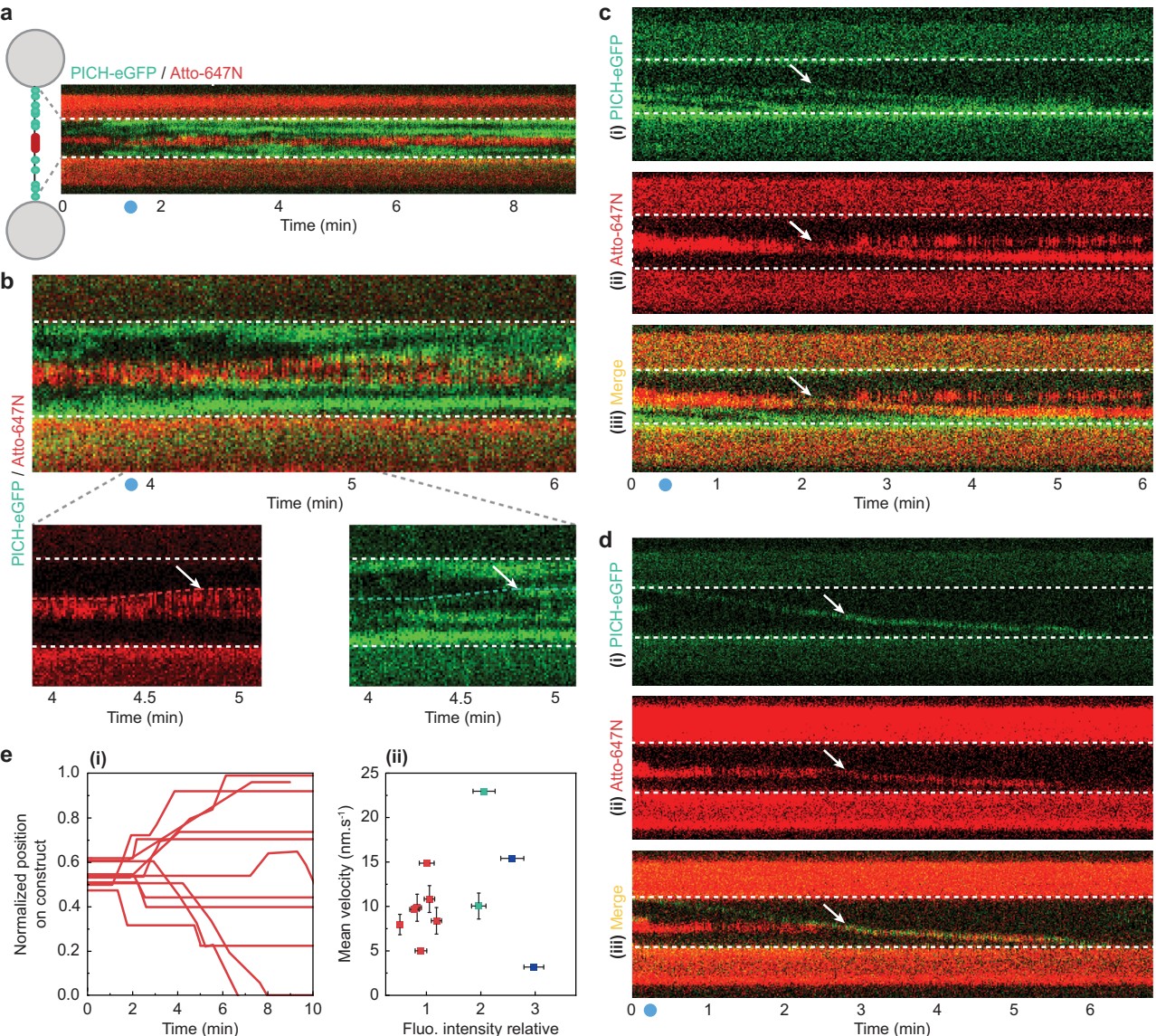

**Fig. 4 | PICH can slide histones along the DNA. a** Schematic representation (*left*) and representative kymograph (*right*) of a nucleosome-array construct with Atto-647N-labelled histones (red), at 10 pN in a microfluidic channel containing PICH-eGFP (green) and ATP. The region of the kymograph depicting the nucleosome-array construct is indicated by the dashed white lines (**b**), *Top*: Representative kymograph of a nucleosome-array construct with Atto-647N-labelled histones (red), at 10 pN in a microfluidic channel containing PICH-eGFP (green) and ATP. *Bottom*: A zoom-in of the Atto-647N signal (*left*) and PICH-eGFP signal (*right*) highlights histone displacement. Histone displacement (red dashed line) co-localizes with PICH-eGFP translocation (green dashed line). White arrows indicate collision of the co-translocating PICH and histones with additional PICH protein. **c** Representative kymographs at 12.5 pN, showing PICH-eGFP (green) (**i**), Atto647N labelled histones (red) (**ii**), and a composite of the two (**iii**). In these kymographs, a PICH-eGFP monomer co-localizes with Atto-647N signal at the start of the measurement (t = 0) and initiates histone sliding (white arrow). **d** Representative kymograph at 12.5 pN with PICH-eGFP (green) (**i**), Atto647N labelled histones (red) (**ii**), and a composite of the two (**iii**), where a PICH-eGFP dimer initiates histone sliding at the instant of collision with the Atto-647N labelled histones. In (**a**–**d**), the light blue dot on the x-axis indicates the moment all nucleosomes had unwrapped, and the height of the dashed white box corresponds to 3 μm. **e** (**i**) Plot showing the trajectories (N = 11) of all histone sliding events along the DNA length, as observed on constructs containing Atto-647N-labelled histones and a low PICH coating at 12.5 pN. **ii** Plot showing the mean velocity during directed translocation (N = 28) as a function of eGFP fluorescence intensity relative to a PICH-eGFP monomer (see Methods). Each data point corresponds to a single trajectory shown in (**i**). Apparent monomers, dimers and trimers are depicted in red, green and purple, respectively. Error bars correspond to SEM. Source data are provided as a Source Data File.

## Discussion

By exploiting the combination of dual-trap optical tweezers, confocal fluorescence microscopy and in vitro reconstituted nucleosome-array constructs, we reveal that the SNF2 protein PICH is a nucleosome remodeler that is force dependent, and thus optimized for its specific cellular role. We demonstrate that PICH can invade nucleosome arrays and use the free energy of ATP hydrolysis to facilitate inner-turn unwrapping at tensions of ≥3 pN. Moreover, we find that, after

nucleosome unwrapping, collision between a translocating PICH and unwrapped histones can result in histone sliding along the DNA.

The observation that PICH is only able to invade nucleosome arrays when they are under tension provides additional evidence for the previous predictions that, in vivo, PICH acts as a (indirect) sensor for chromatin under tension[16]. Thus, our results support the notion that PICH will only bind to sister chromatid entanglements when slightly elevated forces (exerted via the mitotic spindle) locally

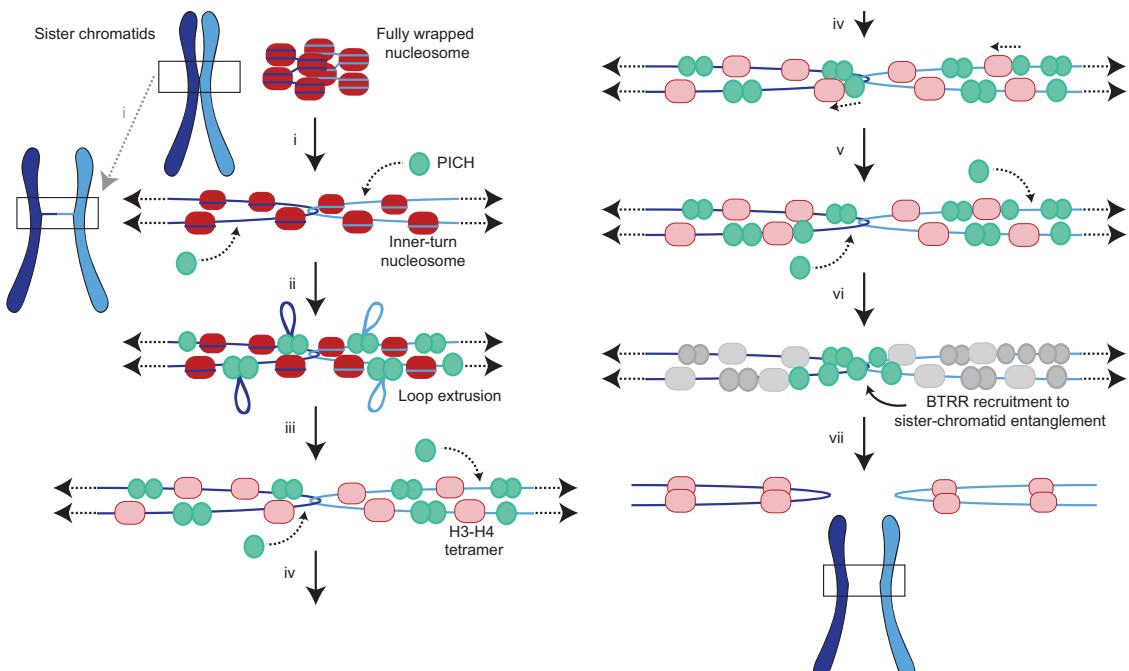

**Fig. 5 | Schematic representation of the proposed model for the role of PICH in UFB dissolution. i** At the early stages of mitosis, when low tensions (represented by black horizontal arrows) are exerted on sister chromatid (purple and light blue) entanglements, nucleosome-nucleosome interactions are disrupted and nucleosome outer turns are unwrapped. This results in nucleosome-free dsDNA regions between neighboring inner-turn nucleosomes. The bare dsDNA regions will be readily bound by PICH (green). Histone octamers are depicted in red. **ii** PICH utilizes ATP hydrolysis to extrude DNA loops. **iii** The ATP-dependent loop extrusion by PICH, and likely the accompanying torsional stress, will facilitate inner-turn unwrapping of the nucleosomes on the UFB, resulting in loss of H2A/H2B dimers and thus, the formation of H3-H4 tetramers (pink). Consequently, an increasing amount of nucleosome-free DNA becomes available for PICH to bind to. **iv** Bound PICH can slide remaining histones (e.g., H3 and/or H4) along the UFB. **v** PICH-induced histone sliding increases the accessibility of the DNA entanglement, allowing additional PICH to bind. **vi** Binding of PICH to the DNA entanglement results in a sufficient recruitment platform for the BTRR dissolvasome and stimulation of TopoIIα (not shown), which, in turn, (**vii**) facilitates the unlinking of the DNA entanglement and, therefore, a timely dissolution of the UFB.

destabilize the chromatin, yielding small regions of nucleosome-free dsDNA. This efficient, early recognition of chromatin bridges by PICH would furthermore provide an explanation for the in vivo observation that, already in early mitosis, PICH can be observed on very short (i.e., weakly stretched) chromatin bridges[26]. Based on our analysis of the probability of inner-turn nucleosome unwrapping and mean nucleosome lifetimes, PICH facilitates unwrapping most efficiently at forces between ~5–10 pN. As forces acting on separating sister chromatids have been reported ranging over three orders of magnitude[56–58], we are unable to compare this observed optimal force range for PICH to forces present on UFBs in vivo. However, our results do raise the possibility that PICH might encounter a force range of ~5–10 pN in vivo.

SNF2 proteins generally display two different mechanisms for the coupling of ATP hydrolysis to nucleosome remodeling: via DNA-translocation induced collision or indirectly through DNA loop extrusion and the accompanying torsional stress[10,11]. While our current assay does not allow us to definitely answer which mechanism PICH uses to unwrap nucleosomes, it is likely that ATP-dependent loop extrusion plays an important role for the following two reasons. First, during the PICH-induced destabilization of nucleosomes under tension, we observed extensive loop extrusion. Second, the finding that the collision-induced sliding of histones typically happened minutes after unwrapping suggests that PICH-induced nucleosome unwrapping is not simply caused by direct contact between PICH and the nucleosome, but is the result of an indirect long-range effect, such as loop extrusion. Therefore, we propose that, in vivo, after binding to the emerging bare dsDNA binding sites on the arising UFB (Fig. 5, (i)), PICH oligomerizes and utilizes ATP hydrolysis to promote nucleosome unwrapping via loop extrusion (Fig. 5, (ii)). Previous studies have shown that nucleosome inner-turn unwrapping coincides with the dissociation of H2A/H2B dimers[49,59,60], which is consistent with the

in vivo observation that PICH-coated UFBs are devoid of histone H2B[26,29,30]. H3 and H4 histones have, however, been found to be more resistant to dissociation from the DNA upon nucleosome unwrapping in vitro[42,49,59]. This chimes with our findings that, even in the presence of PICH and ATP, many of the labelled histones remained bound long after nucleosomes had unwrapped. Together, this raises the possibility that UFBs in vivo may still exhibit bound H3 and/or H4 histone proteins (Fig. 5, (iii)).

Since every nucleosome unwrapping event will yield additional regions of free dsDNA, progressive unwrapping of nucleosomes on a UFB allows increasing numbers of PICH to bind (Fig. 5, (iii)). Consequently, the region of the entanglement (i.e., the catenane) will become bound by PICH, which is then able to slide the remaining histones (e.g., H3 and/or H4) (Fig. 5, (iv)). Histone sliding is generally thought to facilitate exposure of DNA segments[5,7]. Consequently, we postulate that the function of histone sliding by PICH is to enhance catenane accessibility, as has also been previously hypothesized by Ke et al.[30]. This would allow additional PICH proteins to interact with the catenane (Fig. 5, (v)), which, in turn, enables stimulation of TopoIIα and efficient recruitment of the BTRR dissolvasome localized at the DNA entanglement (Fig. 5, (vi)). Both of these processes are thought to play a key role in the timely dissolution of UFBs[17,26] (Fig. 5, (vii)).

Our results have characterized PICH as a tension-dependent nucleosome remodeler that is uniquely adapted for initiating the dissolution of UFBs in vivo. More generally, our findings may have implications for the study of related remodelers. A two-step remodeling mechanism of loop extrusion-induced unwrapping followed by collision-induced histone sliding might also apply to other CSB/ERCC6 proteins. We note that a similar nucleosome remodeling mechanism was proposed for the subfamily member CSB, in which CSB actively wraps and unwraps DNA to remodel nucleosomes[22]. Furthermore, CSB

was found to directly interact with histones[18], which would be consistent with histone-sliding activity. Although direct evidence supporting our proposed two-step mechanism has not yet been found for related proteins, the versatile assay presented here could be used to study chromatin remodeling by other SNF2 proteins at the single-molecule level. We anticipate that future work will seek to unravel the similarities and differences between related SNF2 proteins, highlighting how their function is specialized for their distinct cellular roles.

## Methods

### Reagents

All chemicals were purchased from Sigma-Aldrich. Plasmids were purified using the QIAprep Spin Miniprep kit (Qiagen). Restriction digestion and biotinylation were performed in NEbuffer 3.1 (New England Biolabs). The restriction enzyme BglII was purchased from New England Biolabs. Klenow Fragment (3′−5′ exo-), biotin-14-dATP, biotin-14-dCTP, dTTP and dGTP were purchased from Thermo Fisher Scientific. Double stranded competitor DNA of 147 base pairs was purchased from Integrated DNA Technologies (for sequence, see ref. [42]). EDTA-free protease inhibitor tablets were purchased from Roche. Recombinant Human Histone Octamers (EpiCypher) were used for the reconstitution of nucleosomes. To visualize histones on the DNA, either recombinant monoclonal rabbit histone H3 antibody conjugated with Alexa Fluor 647 (referred to here as Anti-H3-Alexa647; RRID: AB_2663072, Lot #: VC294388, Thermo Fisher Scientific) or Atto-647N NHS ester (ATTO-TEC) was used.

### Preparation of nucleosome-array constructs

An *Escherichia coli* expression vector was used to express a pKYB1 plasmid containing 12 × 601 repeats. For plasmid construction, see Spakman et al.[42]. The plasmid was linearized by BglII and biotinylated using previously established protocols[61]. The restriction enzyme BglII ensured that the nucleosome positioning sites are located roughly in the centre of the construct (flanked by dsDNA handles of ~4 kilo base pairs and ~4.5 kilo base pairs, respectively). Nucleosome reconstitutions were performed by gradient salt dialysis using a peristaltic pump (Watson Marlow) and a 6–8 kD MWCA micro dialysis system (Hampton Research), following a procedure described by ref. [62]. Reconstitution was performed using different Human Histone Octamer:601 motif ratios, 116 ng/μL of the 12 × 601-pKYB1 plasmid (~200 nM 601 motifs), and 19 ng/μL of competitor dsDNA (~200 nM)[42]. Reconstituted nucleosome-array constructs were stored at 4 °C and were stable for at least 7 weeks.

### Preparation of recombinant PICH-eGFP

Recombinant human PICH-eGFP was expressed in Sf21 insect cells using pFastback-based baculoviruses and subsequently purified, essentially as described previously[16,17]. In brief, cells were infected for ~60 h, after which they were harvested at 4 °C and stored at −80 °C. For cell lysis, the pellet was resuspended on ice in lysis buffer (50 mM Tris-HCl pH 7.5, 500 mM NaCl, 10% glycerol, 50 mM Imidazole, 1 mM PMSF, 1 protease inhibitor tablet (PI)) and cells were dounce homogenized and sonicated. In order to remove cell debris, the lysate was cleared by centrifugation at 20,000 g for 1 h. The lysate supernatant was passed through a 1 mL HisTrap HP affinity column. Unbound protein was washed-out using three different wash buffers (WB) with varying salt concentrations (WB1–WB3). WB1 includes lysis buffer without PMSF and PI; WB2 includes WB1 with 1 M NaCl; and WB3 includes WB1 with 200 mM NaCl. Bound proteins were eluted (using WB3 with 500 mM imidazole) and fractions with PICH-eGFP were pooled and loaded onto a 1 mL HiTrap Heparin HP column. Protein was eluted using a linear salt gradient. Again, fractions with PICH-eGFP were pooled and, subsequently subjected to gel filtration chromatography on a 120 mL HiLoad 16/600 Superdex 200 column equilibrated in buffer (50 mM

Tris-HCl pH 7.5, 200 mM NaCl, 10% glycerol, 0.1 mM EDTA, 1 mM DTT). The appropriate fractions were pooled and dialyzed against storage buffer (50 mM Tris-HCl pH 7.5, 200 mM NaCl, 20% glycerol, 0.1 mM EDTA, 1 mM DTT) for 6 h. The dialyzed protein was centrifuged at 20,000 g for an hour before snap freezing in dry ice and was stored at −80 °C in aliquots (12.5 μL of 1 μM). All the steps were done at 4 °C, unless otherwise stated.

### Labelling of histones on nucleosome-array constructs

To verify the location of nucleosomes on the DNA construct, ~0.7 nM nucleosome-array construct was incubated with 200 μg/mL Anti-H3-Alexa647 (1:5 dilution of 1 mg/mL stock) for 3 h in measurement buffer (see below) at 4 °C. As the labelling efficiency using Anti-H3-Alexa647 was low, for constant-force experiments we switched to labelling the histones on our construct with Atto-647N using NHS ester labelling[42], which yields a much higher degree of labelling. Here, ~0.7 nM nucleosome-array construct was incubated with 200 μM Atto-647N NHS ester (1:10 dilution of 2 mM stock) for ~1 h at room temperature. The incubation buffer consisted of 20 mM Hepes-NaOH pH 7.5, 100 mM NaCl, 2 mM MgCl$_2$, 0.02% (v/v) Tween-20 and 0.2% (w/v) BSA.

### Single-molecule experiments

A commercially available combined dual-trap optical tweezers and confocal fluorescence microscopy instrument (C-trap, LUMICKS B.V.) was used. The instrument was controlled using a custom-written program in LabVIEW 2011. Experiments were performed in a multi-channel laminar flow cell containing 5 separate channels (Supplementary Fig. 1). Channels 1 and 2 were used to optically trap streptavidin-coated polystyrene beads (1.76 μm diameter, Spherotech BV) and tether biotinylated nucleosome-array constructs, respectively[63,64]. This dumbbell-like construct could then be rapidly moved, via channel 3, to either channel 4 or 5[63–65]. All channels contained the measurement buffer, which consists of 20 mM Hepes-NaOH (pH 7.5), 100 mM NaCl, 2 mM MgCl$_2$, 0.02% (v/v) Tween-20, 0.2% (w/v) casein, 0.1 mM EDTA and 1 mM sodium azide. The presence of 0.2% casein in the measurement buffer ensured minimal dissociation/degradation of the nucleosomes from the DNA template, as previously shown for the blocking agent BSA[42,66]. Channels 4 and 5 were used to study nucleosome unwrapping in the absence and presence of PICH, respectively. To this end, the buffer in channel 5 additionally contained 25 nM PICH-eGFP, 2 mM ATP and 1 mM DTT, unless stated otherwise. For studying PICH-induced histone sliding, Atto-647N labelled nucleosome-array constructs were briefly incubated in the channel containing PICH (at a tension of 3 pN) until several PICH proteins had bound (as determined by visual inspection of the fluorescence images). These nucleosome-array constructs were then moved to channel 4, which in this case, was supplemented with 2 mM ATP and 1 mM DTT. The Force-Distance data were acquired at 50 Hz in all experiments. Data were obtained at room temperature.

### Fluorescence imaging

Fluorescence snapshots of the tethered nucleosome-array construct were generated by consecutive confocal scans along a line to generate a confocal image of 39 × 144 pixels. A pixel size representing 75 nm in the sample was used for all fluorescence imaging experiments. Kymographs were created by means of confocal line scans between the center of the two beads at each time point. Imaging settings for visualizing PICH-eGFP and histones (labelled with either Anti-H3-Alexa647 or Atto-647N) are summarized in Supplementary Table 1. Together, these settings allowed detection of single PICH-eGFP proteins, as determined by single photo-bleaching steps and blinking of the fluorophore. Since Atto-647N NHS ester is a non-specific dye that can covalently bind to any free amine in a protein, the intensity of the Atto-647N fluorescence signal could not be used to determine the number of histones on the nucleosome-array construct. Therefore, in

our analysis, the Atto-647N fluorescence signal is only used as a probe for the presence of histones.

## Identification and characterization of inner-turn nucleosome unwrapping events

Distance-Time and Force-Distance curves were extracted using a custom-written LabVIEW program. A previously described MatLab-based (ML 2016b) step-fitting algorithm that detects horizontal segments in graphs[67,68] was used to identify inner-turn nucleosome unwrapping events and obtain nucleosome lifetimes and step sizes from Distance-Time traces (obtained during constant-force measurements). All unwrapping events identified by the algorithm were visually verified by monitoring the end-to-end length of the nucleosome arrays (an increase in length of ~26 nm should be observed for each unwrapping event). To exclude transient PICH-induced shortening events, likely caused by loop-extrusion activity[34], only steps that resulted in an overall increase in length were included in the analysis. The duration between the start of the constant-force measurement and the start of each horizontal segment determined the lifetime of a single nucleosome on the array. The vertical distance between two horizontal segments determined the step size. Step sizes of ≥40 nm were assumed to reflect double steps, where the inner turns of two nucleosomes are unwrapped simultaneously. Therefore, the associated lifetimes of these steps were counted twice. Step sizes indicating that three or more steps occurred simultaneously were not encountered (Supplementary Fig. 3). The obtained nucleosome lifetimes for each condition were subsequently fitted to a bi-exponential decay function using OriginPro 2019. To determine the number of nucleosomes that remained wrapped during the constant-force measurements, the Force-Distance curves (obtained by stretching the nucleosome-array construct up to at least 35 pN, after the constant-force measurement) were visually inspected to identify additional force-ruptures and ~26 nm shifts in DNA contour length, associated with inner-turn nucleosome unwrapping[2,42]. The number of nucleosomes that unwrapped under constant tension and those that remained wrapped were used to calculate the fraction of unwrapped nucleosomes (Eq. 1). To correct this fraction for the presence of tetrasomes, we subtracted the observed tetrasome population (of ~26%) from the previously determined unwrapped fractions α, using the following equation, $\alpha_{corrected} = (\alpha - 0.26)/(1 - 0.26)$. A Python (3.7) algorithm was used to apply a logistic regression analysis to the unwrapped fractions in the absence or presence of PICH and ATP. This resulted in sigmoidal curves that approximate the force dependency of the probability of inner-turn nucleosome unwrapping. All errors represent a standard error of the mean (SEM), unless indicated otherwise. 95% confidence intervals (CIs) for the proportions and probabilities of nucleosomes that unwrapped during the constant force measurement were calculated using the Clopper-Pearson method[69,70].

## Fluorescence image analysis

In order to extract the localization pattern of PICH on the nucleosome-array constructs, intensity profiles were made by measuring the background-corrected photon counts over a line scan between the two beads in the kymograph, averaged over the first 6 s of the constant-force measurement. To determine the coverage of PICH proteins on a nucleosome-array construct, and their oligomeric states, a Gaussian function was fitted to the average background-corrected intensity profile of a single PICH-eGFP monomer. For this, ≥50 intensity profiles originating from ≥5 different PICH-eGFP monomers were used for each set of imaging settings (Supplementary Table 1). The Gaussian-fitting was performed using OriginPro 2019. The obtained Gaussian distribution corresponds to the expected point spread function (PSF) of a monomer and was used to determine the Gaussian distributions of PSFs of apparent PICH-eGFP oligomers (up to pentamers). Next, for each nucleosome-array construct, background-corrected intensity

profiles over line scans of the first 10 s of constant-force measurements were averaged. Then, by using the Peak Analyzer tool of OriginPro 2019, peaks of the obtained intensity profiles were fitted using width and area constraints derived from the Gaussian distributions of the PSFs of PICH-eGFP monomers and 'apparent' oligomers. Note that due to the diffraction-limited resolution of our fluorescence imaging setup, we cannot distinguish between true oligomers and monomers that are sufficiently close that their intensity peaks fully overlap. However, for our purposes, it is not essential to differentiate between true oligomers and neighboring PICH monomers. For this reason, sufficiently close PICH monomers are referred to as apparent oligomers. To determine the apparent oligomeric state of bound PICH-eGFP that is responsible for histone sliding, the integrated density of background-corrected intensity profiles (at $N = 5$ or $N = 10$ time points, depending on the length of the track) of these proteins were compared to the area under the curve of the PSF of PICH-eGFP monomers. Within all timescales where PICH-eGFP coating and oligomeric states were determined, the effect of photo-bleaching was negligible, as experimentally determined from individual bleaching traces. For estimating the average velocity of PICH-mediated histone sliding, trajectories of translocating PICH-eGFP and co-localized Atto-647N-labelled histones were manually tracked using the open-source software ImageJ (v1.53c). Segments of tracks with a velocity of <3 nm s$^{-1}$ were defined as pauses, whereas segments with a velocity of ≥3 nm s$^{-1}$ were defined as active translocation. All active translocation segments with a duration of ≥3 s were included in average velocity calculations. All errors represent an SEM.

## Reporting summary

Further information on research design is available in the Nature Portfolio Reporting Summary linked to this article.

## Data availability

Source data are provided with this paper. In addition, all data generated in this study have been deposited in the Dataverse repository at https://doi.org/10.34894/RDQLB5.

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

## Acknowledgements
We thank Dr Anna H. Bizard for useful comments on the paper. This project has received funding from the Netherlands Organization for Scientific Research (NWO), via a Chemical Sciences TOP grant [714.015.002] (to G.J.L.W., E.J.G.P., G.A.K), the European Research Council (ERC) under the European Union's Horizon 2020 research and innovation programme [883240] Monochrome (to G.J.L.W.), the Novo Nordisk Foundation [NNF18OC0034948] (to I.D.H and G.J.L.W.), and the European Union Horizon 2020 grants Chromavision [665233] (to G.J.L.W., I.D.H. and E.J.G.P.). M.I.S and I.D.H. were funded by the Danish National Research Foundation [DNRF115] and the Nordea Foundation.

## Author contributions
D.S. and A.S.B. designed the experiments. D.S. and T.V.M.C. performed the experiments. D.S. and T.V.M.C analyzed the data. D.S. and T.V.M.C wrote the draft paper. M.I.S. purified recombinant PICH-eGFP. All authors discussed the data and edited the paper. G.A.K., I.D.H., E.J.G.P. and G.J.L.W. supervised the research.

## Competing interests
The combined optical tweezers and fluorescence technologies used in this paper are patented and licensed to LUMICKS B.V., in which E.J.G.P. and G.J.L.W. have a financial interest. The remaining co-authors have no competing interests with the content of this paper.
