## [Peer Review File · Nature Communications]

REVIEWER COMMENTS

Reviewer #1 (Remarks to the Author):

In this manuscript, Spakman et al. used optical tweezers combined with confocal fluorescence microscopy to characterize PICH as a tension-dependent nucleosome remodeler. This unusual and interesting activity may underlie the role of PICH in chromatin-bridge dissolution. The data and analyses are in general convincing and of high quality. However, the following points need to be addressed before the paper is acceptable for publication.

(1) The authors should demonstrate that the lack of PICH-nucleosome binding at 1 pN in Fig. 1b is not due to the antibody labeling of histones, which could obstruct PICH interaction.

(2) The loop extrusion activity of PICH inferred from Fig. 2b should be shown on a bare DNA substrate, which would also allow the authors to better characterize its force dependence.

(3) The authors attribute the fast unwrapping population in Fig. 3 to tetrasomes. While this is a reasonable speculation, a proof would need experiments using tetrasome-loaded DNA, or the conclusion should be toned down.

(4) For the histone sliding results in Fig. 4, can the authors show or discuss how many nucleosomes in the array are mobilized based on the fluorescence signal? Presumably the ones on the periphery tend to be moved first? Since all histones are labeled in this experiment, what is the likelihood that only H2A-H2B dimers are mobilized? A mononucleosome substrate would help clarify these points.

(5) The authors should discuss the similarities and differences in the magnitude and direction of force applied in the single-molecule experiments versus that existing in anaphase chromatin bridges in the context of Fig. 5.

Reviewer #2 (Remarks to the Author):

During anaphase, chromosomes break at the centromere and the resulting chromatids separate and move apart. Any remaining DNA entanglements must be resolved for mitosis to succeed. SNF2 is a subset of the SWI/SNF (switch/sucrose non-fermentable) family of proteins that contain an ATPase activity that destabilizes DNA-histone interactions. Though generally associated with regulation of transcription, the SNF2 protein PICH (also known as ERCC6L) is associated with chromatin during mitosis. The principal effort of this paper is to demonstrate that PICH is capable of remodeling nucleosomes. Intriguingly, though PICH does have an affinity for DNA, it is not known to bind directly to nucleosomes. This work proposes that PICH must bind to and move along DNA, disrupting nucleosomes along the way, facilitating the ultimate breakup of chromatin bridges between chromatids.

Building on a previous study that demonstrated both that PICH has an affinity for DNA and will diffuse along the double strand, this work combines single molecule force measurements and time resolved imaging (kymographs) on prepared, labeled arrays of nucleosomes flanked by long handles of DNA. Force spectroscopy experiments reveal that PICH will destabilize nucleosomes, though only in the presence of ATP. Furthermore, fluorescence imaging convincingly shows that PICH binds preferentially away from the nucleosome (along the handles or the linking regions between neighboring nucleosomes) and after diffusion along the DNA, collides with and disrupts the nucleosome. Surprisingly,

PICH appears to slide along with the remaining tetramer. PICH is also observed to generate DNA loops.

These are compelling results that build upon a range of previous studies and provide new insights into the interesting role of this protein. The experiments have been carefully carried out, are clearly described, and present a complete story. This work should be published after addressing a few specific questions.

Specific Questions:

Figure 1: Unwrapping is shown over a range of forces. What is the corresponding released length of DNA during these events? Was this length change affected by the applied force?

Figure 2: PICH-mediated DNA looping is hypothesized to explain transient drops in the measured length. However, this looping is not characterized. Drops in length are interpreted as loops. However, how can this be distinguished from simply rewinding of nucleosomes? Additional discussion about this would be helpful.

Page 7, Page 9 and Figure 4: Though PICH-histone movement is well discussed, there must be some likelihood of PICH-histone dissociation, even in the presence of casein. What fraction of trajectories showed dissociation?

Page 8 and Figure 4: Why were two different dyes used to label H3? If Atto-647N was used to track movement, but could not be used to reliably count the number of tetramers, was Alexa Fluor 647 used for this purpose? If not, is there any information on the number of tetramers being pushed around by PICH?

Page 8 and Figure 4: The green and blue is difficult to tell apart in e(ii).

Page 9: Could a reference for the effect of casein on nucleosomes (or control experiments) be provided?

Supplementary Figure 2: The force distributions appear to have a horizontal spread. Is this real and due to an error in the force clamp? Does this imply some uncertainty in the force?

Supplementary Figure 2: The measured step sizes have a considerable range – were double steps corresponding to two nucleosomes disrupted simultaneously ever seen or were they removed from the data?

Reviewer #3 (Remarks to the Author):

The manuscript entitled “PICH acts as a force-dependent nucleosome remodeler” presented an in-vitro experiment on elucidating the force-dependent role of SNF2 protein PICH in unwrapping and sliding nucleosomes. Spakman et al. systematically studied how PICH invades stretched nucleosome array and mediates force-induced nucleosome unwrapping. The authors observed how PICH + ATP helps lower the force required for unwrapping the inner turn of nucleosomes by estimating the unwrapped fraction in a given window of time and measuring nucleosome lifetimes at different forces (10-15 pN). Using fluorescent assay, the authors also showed histone sliding due to the collision of PICH proteins and co-localized histones on the bare DNA.

The manuscript is well-written and of suitable quality and general interest to justify its publication, after revision to address the following comments:

1. The authors may need to distinguish PICH's weak chromatin remodeling and potential strong DNA translocation activity. The former is caused by the poor accessibility to DNA around the nucleosomes due to nucleosomal array compaction [Fig. 1.9 in Zhang, Y. et al. *Methods Enzymol.* 513, 3-28 (2012)], while the latter is not well explored in this work. Previous single-molecule experiments based on optical tweezers demonstrated the chromatin remodellers contain strong ATPases that can translocate DNA against force up to 40 pN with a processivity of 35 bp and speed of 25 bp/s [Zhang, Y.,..., Bustamante, C. *Mol Cell*, 24, 559-568 (2006); Sirinakis, G. et al. *EMBO J*, 30, 2364-2372 (2011)]. The DNA translocation properties of PICH appear to be consistent with the ATPases in other chromatin remodellers and should be compared with citations of these earlier highly related papers.

2. Line 155: The authors increase the concentration of PICH by at least four-fold to achieve a similar coverage of PICH on the nucleosome-array construct to elucidate the dependence of ATP hydrolysis on inner-turn unwrapping of nucleosome under tension. Is PICH binding to DNA dependent upon ATP?

3. Lines 166-167: I am confused with the definition of nucleosome lifetime. How is the lifetime of nucleosomes measured on an array of nucleosomes related to the lifetime of a single nucleosome?

4. Lines 231-232: How is "maximum activity" defined? For example, is it in the force range where the unwrapping probability is greater than some particular value? If yes, how is that "value" determined?

5. Figures 3b and c show two timescales, corresponding to slow and fast unwrapping events. Can reducing the tension (e.g., 10 pN to 7.5 pN) lead to the re-wrapping of the unwrapped nucleosomes? Is it a reversible process?

6. Does the unwrapping process have any effect on the histone core? If yes, what would be its implications?

Reviewer #1 (Remarks to the Author):

In this manuscript, Spakman et al. used optical tweezers combined with confocal fluorescence microscopy to characterize PICH as a tension-dependent nucleosome remodeler. This unusual and interesting activity may underlie the role of PICH in chromatin-bridge dissolution. The data and analyses are in general convincing and of high quality. However, the following points need to be addressed before the paper is acceptable for publication.

(1) The authors should demonstrate that the lack of PICH-nucleosome binding at 1 pN in Fig. 1b is not due to the antibody labeling of histones, which could obstruct PICH interaction.

We thank the reviewer for highlighting this point. Only 4 out of the 14 nucleosome constructs used to create the lower panel of Figure 1b (i) contained antibody-labelled histones (due to the low labelling efficiency of this method). To exclude that these 4 cases skewed our results, we have added a new Supplementary Figure 2, in which we have split up the results from this panel into labelled and unlabelled constructs. This new figure shows that the lack of PICH binding to the nucleosome area does not depend on the presence or absence of antibodies on the histones. We have highlighted this in the revised manuscript (lines 114-116) by adding the following statement: "Additionally, by comparing results from labelled and unlabelled constructs, we excluded the possibility that the relative absence of PICH in the nucleosome array at ~1 pN was merely due to the anti-H3-Alexa647 antibody blocking its access (Supplementary Fig. 2)."

(2) The loop extrusion activity of PICH inferred from Fig. 2b should be shown on a bare DNA substrate, which would also allow the authors to better characterize its force dependence.

We would like to point out that evidence for DNA loop extrusion activity of PICH has been shown by Bizard et al. 2019 (reference 34). In that magnetic tweezers study, PICH-mediated loop extrusion on bare DNA was characterized at forces up to 8 pN. Therefore, additional characterization of PICH-mediated loop extrusion was not needed to support our findings. Nonetheless, we agree with the referee that it may not have been sufficiently clear from our text that the loop extrusion activity of PICH has already been characterized. Therefore, we now emphasize this in lines 137-139 of the main text as follows: "These events occurred throughout the constant-force measurements, independent of the presence of nucleosomes, and are indicative of the loop-extrusion activity of PICH that has previously been reported on bare DNA under tension by Bizard et al."

(3) The authors attribute the fast unwrapping population in Fig. 3 to tetrasomes. While this is a reasonable speculation, a proof would need experiments using tetrasome-loaded DNA, or the conclusion should be toned down.

The reviewer rightly indicates that our conclusions regarding the presence of tetrasomes on our construct is not based on direct evidence. We therefore changed the concluding sentence of this paragraph (lines 195-197) to the following: "In line with these earlier observations, we speculate that the minor, faster population could reflect the unwrapping of tetrasomes."

(4) For the histone sliding results in Fig. 4, can the authors show or discuss how many nucleosomes in the array are mobilized based on the fluorescence signal (1)? Presumably the ones on the periphery tend to be moved first (2)? Since all histones are labeled in this experiment, what is the likelihood that only H2A-H2B dimers are mobilized (3)? A mononucleosome substrate would help clarify these points. We thank the reviewer for these interesting questions. Below we will answer these three questions separately.

1) Due to the nature of the unspecific protein labelling using the Atto-647N NHS ester, the number of fluorescently-labelled histones on each construct and the number of fluorophores present per labelled histone protein varies and cannot be determined. Therefore, we cannot draw conclusions on how

many nucleosomes per constructs were mobilized. This is stated in the methods section (lines 436-440): “Since Atto-647N NHS ester is a non-specific dye that can covalently bind to any free amine in a protein, the intensity of the Atto-647N fluorescence signal could not be used to determine the number of histones on the nucleosome-array construct. Therefore, in our analysis, the Atto-647N fluorescence signal is only used as a probe for the presence of histones.” Future studies, using more direct labelling methods such as genetically encoded fluorescent tags, would allow to answer such questions in more detail.

2) As discussed in lines 274-275 of the main text, histone sliding was only observed in cases where clear co-localization of the fluorescence signals of PICH and histones was observed. This indicates that the first histones that are moved would be the histones that are in closest proximity to PICH, which could be either central or peripheral histones. However, as central histones often have nearby roadblocks, sliding over significant distances would be more likely for peripheral histones.

3) It is true that the labelling method used for the histone sliding experiments cannot distinguish between the different histone variants. However, as stated in the discussion section (lines 328-332), we think that PICH-mediated sliding occurs predominantly to histones H3 and H4 for the following reasons. Previous studies have shown that nucleosome inner-turn unwrapping coincides with the dissociation of H2A/H2B dimers, while H3 and H4 histones are more resistant to dissociation from the DNA. This is consistent with the observation that PICH-coated UFBs are devoid of histone H2B, but that histone H3 remains bound to the DNA after force-induced nucleosome unwrapping. For this reason, we speculate that the probability of PICH encountering H3 and H4 histones is larger than encountering H2A and H2B histones. However, we do not have reasons to believe that the histone sliding activity of PICH is specific to certain variants of histones. Also here, more direct labeling methods would allow this question to be addressed.

(5) The authors should discuss the similarities and differences in the magnitude and direction of force applied in the single-molecule experiments versus that existing in anaphase chromatin bridges in the context of Fig. 5.

We are grateful to the reviewer for bringing up this point and agree that this would be a highly relevant comparison. However, to our knowledge, there are currently no studies that report what tensions UFBs are typically subjected to *in vivo*. In fact, reported forces experienced by separating sister chromatids during mitosis vary over three orders of magnitude (see Ferraro-Gideon et al., 2013, for an overview). Furthermore, even knowing the forces present during chromatid separation would not paint the complete picture of the local tension present on a transient structure such as an UFB. We are therefore currently limited to speculations on the *in vivo* force range based on our *in vitro* results. In order to highlight the uncertainty regarding the tension on UFBs *in vivo*, we have added the following statement to lines 313-316 of the revised manuscript:

“As forces acting on separating sister chromatids have been reported ranging over three orders of magnitude⁵⁶⁻⁵⁸, we are unable to compare this observed optimal force range for PICH to forces present on UFBs *in vivo*. However, our results do raise the possibility that PICH might encounter a force range of ~5-10 pN *in vivo*.”

Reviewer #2 (Remarks to the Author):

During anaphase, chromosomes break at the centromere and the resulting chromatids separate and move apart. Any remaining DNA entanglements must be resolved for mitosis to succeed. SNF2 is a subset of the SWI/SNF (switch/sucrose non-fermentable) family of proteins that contain an ATPase activity that destabilizes DNA-histone interactions. Though generally associated with regulation of transcription, the SNF2 protein PICH (also known as ERCC6L) is associated with chromatin during

mitosis. The principal effort of this paper is to demonstrate that PICH is capable of remodeling nucleosomes. Intriguingly, though PICH does have an affinity for DNA, it is not known to bind directly to nucleosomes. This work proposes that PICH must bind to and move along DNA, disrupting nucleosomes along the way, facilitating the ultimate breakup of chromatin bridges between chromatids.

Building on a previous study that demonstrated both that PICH has an affinity for DNA and will diffuse along the double strand, this work combines single molecule force measurements and time resolved imaging (kymographs) on prepared, labeled arrays of nucleosomes flanked by long handles of DNA. Force spectroscopy experiments reveal that PICH will destabilize nucleosomes, though only in the presence of ATP. Furthermore, fluorescence imaging convincingly shows that PICH binds preferentially away from the nucleosome (along the handles or the linking regions between neighboring nucleosomes) and after diffusion along the DNA, collides with and disrupts the nucleosome. Surprisingly, PICH appears to slide along with the remaining tetramer. PICH is also observed to generate DNA loops.

These are compelling results that build upon a range of previous studies and provide new insights into the interesting role of this protein. The experiments have been carefully carried out, are clearly described, and present a complete story. This work should be published after addressing a few specific questions.

Specific Questions:

1) Figure 1: Unwrapping is shown over a range of forces. What is the corresponding released length of DNA during these events? Was this length change affected by the applied force?

As there is no unwrapping data shown in Figure 1, we assume that the reviewer is referring to Figure 2. We think that these questions regarding DNA release length are already addressed in our Supplementary Figure 3 (previously Supplementary Figure 2). In this figure, we show that, at constant forces between 10 and 15 pN, the step size corresponding to the released length of DNA is unaffected by the applied force, ranging from 24.5 to 27 nm.

2) Figure 2: PICH-mediated DNA looping is hypothesized to explain transient drops in the measured length. However, this looping is not characterized. Drops in length are interpreted as loops. However, how can this be distinguished from simply rewinding of nucleosomes? Additional discussion about this would be helpful.

We thank the reviewer for this suggestion. See also our related response to question 2 from reviewer 1. As shown in Figure 2a, in the absence of PICH, force drops are never observed, making it unlikely that they represent rewinding of nucleosomes. See also our previous publication (Spakman et al., 2020) where we discuss that rewinding of nucleosomes under tension is unlikely. Furthermore, DNA loop extrusion by PICH has been characterized before using magnetic tweezers on a bare DNA substrate by Bizard et al., 2019. The force-drops found on our nucleosome-array constructs are nearly identical to those reported there. We also see these force drops on bare DNA constructs in our optical tweezers setup. However, we have decided not to include these data as it is a topic of ongoing research interest and will be included in a future publication. As also mentioned in the response to question 2 from reviewer 1, we have amended the main text to make it clearer that the loop extrusion activity of PICH on bare DNA under tension has previously been characterized. Lines 137-139 now state:

“These events occurred throughout the constant-force measurements, independent of the presence of nucleosomes, and are indicative of the loop-extrusion activity of PICH that has previously been reported on bare DNA under tension by Bizard et al.”

3) Page 7, Page 9 and Figure 4: Though PICH-histone movement is well discussed, there must be some likelihood of PICH-histone dissociation, even in the presence of casein. What fraction of trajectories showed dissociation?

This question could be interpreted as the dissociation of PICH from a histone, or as the dissociation of the complete PICH-histone complex from the DNA. As we are unsure which of these interpretations was meant by the reviewer, we answer both scenarios here. For this, we looked at the low-PICH conditions (as used for Figure 4c-e), where dissociation events could be observed best, and added the following statement (lines 281-284):

“Dissociation of a PICH-histone complex from the DNA was observed only once in eleven trajectories during the tested timeframe. This is consistent with the strong affinity of PICH for DNA, as observed by Biebricher et al.¹⁶. Clear dissociation of PICH from a histone was not observed in any of the eleven trajectories.”

4) Page 8 and Figure 4: Why were two different dyes used to label H3? If Atto-647N was used to track movement, but could not be used to reliably count the number of tetramers, was Alexa Fluor 647 used for this purpose? If not, is there any information on the number of tetramers being pushed around by PICH?

As stated in the methods section, Anti-H3-Alexa647 was first used to confirm the proper positioning of nucleosomes on our construct. However, the low labelling efficiency achieved using Anti-H3-Alexa647 did not enable more high-throughput constant force experiments. Therefore, we switched to using Atto-647N NHS ester for subsequent experiments. This label binds nonspecifically to the free amines on the histones and has a much higher labelling efficiency than Anti-H3-Alexa647. We acknowledge that the motivation for switching labelling methods could have been explained more clearly. Therefore, lines 403-405 in the methods section of this manuscript now contain the following statement:

“As the labelling efficiency using Anti-H3-Alexa647 was low, for constant-force experiments we switched to labelling the histones on our construct with Atto-647N using NHS ester labelling⁴², which yields a much higher degree of labelling.”

We also agree with the reviewer that it would have been very interesting to determine the number of histones that are moved by PICH. However, as also explained in our response to question 4.1 from reviewer 1, due to the nature of the Atto-647N labelling, it is not possible to determine the number of labelled histones that are moved by PICH.

5) Page 8 and Figure 4: The green and blue is difficult to tell apart in e(ii).

We agree with the reviewer that these colors might be difficult to distinguish in this case. To rectify this, we have changed the blue data points in e (ii) to purple and amended the figure caption to reflect this.

6) Page 9: Could a reference for the effect of casein on nucleosomes (or control experiments) be provided?

While we used BSA as an established nucleosome stabilizing agent (Gansen et al., 2007, Li et al., 2012) in a previous publication (Spakman et al., 2020), here, we chose to use casein as a nucleosome stabilizing agent, as casein also prevents non-specific PICH interactions with the surface and DNA (Biebricher et al., 2013). We observed no difference in nucleosome stability between the use of casein in our current study compared with our previous use of BSA to prevent nucleosome dissociation (Spakman et al., 2020). We agree with the reviewer that a comment for the effect of casein on nucleosome stability should have been provided, and therefore revised the statement in lines 418-420 to the following:

“The presence of 0.2% casein in the measurement buffer ensured minimal dissociation/degradation of the nucleosomes from the DNA template, as previously shown for the blocking agent BSA^{42,66}.”

7) Supplementary Figure 2: The force distributions appear to have a horizontal spread. Is this real and due to an error in the force clamp? Does this imply some uncertainty in the force?

The horizontal spread in this figure is a feature of the plotting method (a grouped column scatter plot) to limit overlap and ensure that all datapoints are visible. There is no spread due to uncertainty of the force or an error in the force clamp. We thank the reviewer for pointing out that this way of plotting can be misleading. To avoid potential confusion, we have changed the figure in such a way that the x-axis appears categorical instead of linear.

8) Supplementary Figure 2: The measured step sizes have a considerable range – were double steps corresponding to two nucleosomes disrupted simultaneously ever seen or were they removed from the data?

Double steps were only encountered occasionally in the PICH groups, where nucleosome unwrapping is accelerated. To account for this, steps of over 40 nm were assumed to be double steps, with two nucleosomes unwrapping simultaneously. For the nucleosome lifetime calculations, the time points at which these double steps occurred were counted twice. To make this clear to the reader, we have added the following in the method section of this manuscript (lines 452-455):

“Step sizes of ≥ 40 nm were assumed to reflect double steps, where the inner turns of two nucleosomes are unwrapped simultaneously. Therefore, the associated lifetimes of these steps were counted twice. Step sizes indicating that three or more steps occurred simultaneously were not encountered (Supplementary Fig. 3).”

Reviewer #3 (Remarks to the Author):

The manuscript entitled “PICH acts as a force-dependent nucleosome remodeler” presented an in-vitro experiment on elucidating the force-dependent role of SNF2 protein PICH in unwrapping and sliding nucleosomes. Spakman et al. systematically studied how PICH invades stretched nucleosome array and mediates force-induced nucleosome unwrapping. The authors observed how PICH + ATP helps lower the force required for unwrapping the inner turn of nucleosomes by estimating the unwrapped fraction in a given window of time and measuring nucleosome lifetimes at different forces (10-15 pN). Using fluorescent assay, the authors also showed histone sliding due to the collision of PICH proteins and co-localized histones on the bare DNA. The manuscript is well-written and of suitable quality and general interest to justify its publication, after revision to address the following comments:

1. The authors may need to distinguish PICH’s weak chromatin remodeling and potential strong DNA translocation activity. The former is caused by the poor accessibility to DNA around the nucleosomes due to nucleosomal array compaction [Fig. 1.9 in Zhang, Y. et al. *Methods Enzymol.* 513, 3-28 (2012)], while the latter is not well explored in this work. Previous single-molecule experiments based on optical tweezers demonstrated the chromatin remodellers contain strong ATPases that can translocate DNA against force up to 40 pN with a processivity of 35 bp and speed of 25 bp/s [Zhang, Y.,..., Bustamante, C. *Mol Cell*, 24, 559-568 (2006); Sirinakis, G. et al. *EMBO J*, 30, 2364-2372 (2011)]. The DNA translocation properties of PICH appear to be consistent with the ATPases in other chromatin remodellers and should be compared with citations of these earlier highly related papers.

We agree that the DNA translocation activity of PICH is not thoroughly explored here. However, this activity of PICH was the main focus of a previous publication from our group (Biebricher et al., 2013) using similar experimental approaches as used in this work. In the previous study, PICH translocation activity was extensively characterized and compared to other SNF2 family members. In particular, it was shown that contrary to other SNF2 remodelers, PICH was unable to efficiently induce nucleosome unwrapping in the absence of tension on the DNA, while it did show processive, but reversible translocation. Thus, for detailed information regarding the DNA translocation activity of PICH we would like to refer the reviewer to this publication. Nonetheless, we thank the reviewer for pointing out that we did not yet refer to the highly relevant papers regarding DNA translocation activities of

other SNF2 proteins mentioned above. In the revised manuscript we cite these works in an amended sentence in the introduction (lines 63-66) where we discuss similarities and differences between PICH and other SNF2 remodelers (with references 34-36 corresponding to the works suggested by the reviewer):

“Although the DNA translocation and looping properties of PICH are similar to those of other SNF2 remodelers^{16,33-36}, its nucleosome remodeling activity has been reported to be orders of magnitude less efficient than that of other SNF2 remodelers¹⁶.”

2. Line 155: The authors increase the concentration of PICH by at least four-fold to achieve a similar coverage of PICH on the nucleosome-array construct to elucidate the dependence of ATP hydrolysis on inner-turn unwrapping of nucleosome under tension. Is PICH binding to DNA dependent upon ATP? We agree with the reviewer that our results indicate that PICH binding to DNA is promoted by the presence of ATP, at least under the conditions employed for *in vitro* single molecule experiments. This finding is also consistent with previous single molecule work, where ATP was always present in the buffer used to load PICH onto the DNA (Biebricher et al., 2013). In order to highlight this finding, we now added the following statement to the main text where we discuss this in line 161: “[...] indicating a stimulatory effect of ATP on PICH binding to DNA”

3. Lines 166-167: I am confused with the definition of nucleosome lifetime. How is the lifetime of nucleosomes measured on an array of nucleosomes related to the lifetime of a single nucleosome? In this manuscript, we defined the nucleosome lifetime as “the duration that each nucleosome remains wrapped within a 10-minute period at a constant force”. With this definition, we refer to an individual nucleosome on our nucleosome array. We measure these individual nucleosome lifetimes using a step-fitting algorithm, as described in the methods section under “Identification and characterization of inner-turn nucleosome unwrapping events”. To further clarify that we are indeed measuring lifetimes of single nucleosomes, we changed the sentence in line 172 to the following: “We define this lifetime as the duration that an individual nucleosome on the array remains wrapped during a 10-minute period at a constant force” Furthermore, we revised the section in the methods section that explains how lifetimes are determined (see lines 450-451) to the following: “The duration between the start of the constant-force measurement and the start of each horizontal segment determined the lifetime of a single nucleosome on the array.”

4. Lines 231-232: How is “maximum activity” defined? For example, is it in the force range where the unwrapping probability is greater than some particular value? If yes, how is that “value” determined? We acknowledge that the term “maximum activity” is somewhat ambiguous. Therefore, in the results section where we discuss the increase in unwrapping probability in the presence of PICH (lines 232-234, Figure 3f), we added the following statement to clarify this definition: “The force range within the full width at half maximum (FWHM) of this curve indicates that PICH facilitates unwrapping of nucleosomes most efficiently between 5.3 and 10.7 pN.”

5. Figures 3b and c show two timescales, corresponding to slow and fast unwrapping events. Can reducing the tension (e.g., 10 pN to 7.5 pN) lead to the re-wrapping of the unwrapped nucleosomes? Is it a reversible process? We are grateful to the reviewer for raising these questions. We would like to clarify that the time scale in Figure 3c (iii) (15 pN) differs from the scale in Figure 3b and 3c (i-ii) (10 and 12.5 pN), not so much to distinguish between slow and fast unwrapping events, but to highlight the difference between the PICH and no-PICH condition. Since nucleosome unwrapping at 15 pN occurs much faster than at 10 and 12.5 pN, the difference between the PICH and no PICH condition is not visible at the scale used for the other panels. Regarding nucleosome rewrapping, we would like to refer the reviewer to Brower-Toland et al., 2002, and our previous publication, Spakman et al., 2020. These publications

indeed indicate that reducing the force after nucleosome unwrapping can lead to re-wrapping. Only when higher tensions (>50 pN) come into play, the probability of rewrapping of nucleosomes decreases significantly.

6. Does the unwrapping process have any effect on the histone core? If yes, what would be its implications?

As addressed in our response to question 4.3 from reviewer 1, and highlighted in lines 328-332 in the discussion section of this manuscript, previous studies have shown that nucleosome inner-turn unwrapping coincides with the dissociation of H2A/H2B dimers. We assume that PICH-induced unwrapping of nucleosomes under tension does not differ in its effects from regular force-induced unwrapping, and thus that it coincides with H2A/H2B dimer dissociation.

REVIEWERS' COMMENTS

Reviewer #1 (Remarks to the Author):

The authors have addressed my concerns. I now support publication of the manuscript in the current form.

Reviewer #2 (Remarks to the Author):

The authors have done an excellent job responding to the reviews and we believe the manuscript is now suitable for publication.

Reviewer #3 (Remarks to the Author):

The authors have carefully revised the manuscript, which well addressed my comments. I support the publication of this revised version.